# Sterility Mosaic Disease of Pigeonpea (*Cajanus cajan* (L.) Huth): Current Status, Disease Management Strategies, and Future Prospects

**DOI:** 10.3390/plants13152146

**Published:** 2024-08-02

**Authors:** B. R. Sayiprathap, A. K. Patibanda, Muttappagol Mantesh, Shridhar Hiremath, N. Sagar, C. N. Lakshminarayana Reddy, C. R. Jahir Basha, S. E. Diwakar Reddy, M. Kasi Rao, R. M. Nair, H. K. Sudini

**Affiliations:** 1International Crops Research Institute for the Semi-Arid Tropics, Patancheru, Hyderabad 502324, Telangana, India; 2World Vegetable Center, South and Central Asia, ICRISAT Campus, Patancheru, Hyderabad 502324, Telangana, India; ramakrishnan.nair@worldveg.org; 3Department of Plant Pathology, Acharya NG Ranga Agricultural University, Lam, Guntur 522034, Andhra Pradesh, India; anilpatibanda@yahoo.co.in; 4Department of Plant Pathology, College of Agriculture, University of Agricultural Sciences, Gandhi Krishi Vigynan Kendra (GKVK), Bengaluru 560065, Karnataka, Indiacnlreddy@gmail.com (C.N.L.R.); jahir_basha@rediffmail.com (C.R.J.B.); 5CSIR—North East Institute of Science and Technology, Jorhat 785006, Assam, India; 6Department of Plant Pathology, University of Agricultural Sciences, Dharwad 580005, Karnataka, India; 7School of Agriculture, Mohan Babu University, Tirupati 517102, Andhra Pradesh, India

**Keywords:** *Cajanus cajan*, SMD, PPSMV, eriophyid mite, screening, disease management

## Abstract

Pigeonpea (*Cajanus cajan*) is one of the important grain legume crops cultivated in the semi-arid tropics, playing a crucial role in the economic well-being of subsistence farmers. India is the major producer of pigeonpea, accounting for over 75% of the world’s production. Sterility mosaic disease (SMD), caused by *Pigeonpea sterility mosaic virus* (PPSMV) and transmitted by the eriophyid mite (*Aceria cajani*), is a major constraint to pigeonpea cultivation in the Indian subcontinent, leading to potential yield losses of up to 100%. The recent characterization of another *Emaravirus* associated with SMD has further complicated the etiology of this challenging viral disease. This review focuses on critical areas, including the current status of the disease, transmission and host-range, rapid phenotyping techniques, as well as available disease management strategies. The review concludes with insights into the future prospects, offering an overview and direction for further research and management strategies.

## 1. The Crop

Pigeonpea (*Cajanus cajan* (L.) Huth) is an important grain legume crop cultivated by small and marginal farmers in semi-arid tropical regions of Asia, Africa, and the Americas between 25° N and 35° S [1]. Globally, the pigeonpea crop is harvested from 6.03 million hectares (ha) with a production of about 5.32 million tonnes (MT) (Figure 1) [2]. India is the leading producer, contributing to over 75% of global production [2]. Pigeonpea is primarily grown for its grains, containing 20–30% protein, serving as the principal source of dietary protein for over a billion people [3,4,5,6,7]. Pigeonpea is an important subsistence crop adopted by millions of smallholder farmers and grown sole or intercropped with cereals (finger millet, maize, sorghum, pearl millet), legume crops (groundnut, soybean, Indian bean), cotton, chili, etc., under diverse climatic conditions in a rain-fed agricultural system [5,8]. It is a perennial shrub mainly cultivated as an annual and offers cultivars having different durations: extra-early (<130 days), early (131–150 days), mid-early (151–165 days), and medium duration (166–185 days). During the early phase, its slow growth above ground offers very little competition to the main crop; later, its fast-growing nature with a deep, extensive root system enables the crop to yield in arid conditions with very little available moisture in the soil when no other crop can survive. 

The pigeonpea crop has a direct impact on the economic and financial well-being of subsistence farmers in the subcontinent as it is a low-input, rainfed crop that provides economic returns from every part of the plant. Despite a substantial increase in the area and production of pigeonpea over the last two decades, average yields remain low, with a meager 882 kg/ha globally [2] owing to several biotic (Fusarium wilt, sterility mosaic, Phytophthora blight, and pod borer complex) and abiotic stresses (drought, salinity, and water-logging) encountered at different growth stages. Among the biotic stresses, sterility mosaic disease is the major constraint to pigeonpea production in the Indian subcontinent [5,9,10]. This is a cause of concern as crop yields plateau in the face of rising demand and an increase in the number of mouths to feed.

## 2. Sterility Mosaic Disease (SMD)

SMD of pigeonpea was initially reported in 1931 by Mitra from Bihar state of India [11] and has since evolved into a serious problem for pigeonpea cultivation in the Indian subcontinent, resulting in an annual loss of over US$ 300 million [12]. SMD is endemic to India and has subsequently been reported from Bangladesh, Nepal, Thailand [12,13], Myanmar [14], Sri Lanka [15], and China, but it is not known to occur in Africa or the Americas [12,16]. In India, SMD is reported in various states, including Andhra Pradesh, Bihar, Chhattisgarh, Gujarat, Karnataka, Maharashtra, Punjab, Tamil Nadu, Telangana, Uttar Pradesh, and West Bengal [8,17,18,19]. SMD manifests as a yellow mosaic, with bushy pale green plants with excessive vegetative growth, reduced leaf size, stunting, leaf distortion, and partial or complete cessation of reproductive parts [5,8,20,21,22,23,24,25]. Infected plants also show a significant reduction in plant height, number of branches and flowers per plant, and pod length [26].

SMD is referred to as the “Green Plague” of pigeonpea, as the infected plants exhibit lush greenery with excessive vegetative growth without flowers and seed pods. Under certain conditions, it spreads like a plague in an epidemic form [5,27]. The nature and severity of symptoms depend on the genotype and stage of infection [24]. In susceptible genotypes, early-stage infection (<45-day old plant) results in the development of characteristic symptoms and complete cessation of flowering (sterility), while later-stage infection (>45-day old plant) results in delayed symptom expression with mild mosaic on few branches or on the part of the branch, reduced flowering and partial sterility [5,17]. However, in later stage infected plants, new flushes from ratoon crop (severe pruning) exhibit aggressive and clear severe mosaic symptoms [5,24,25,28]. Based on severity, the disease symptoms are categorized into three groups: (i) severe mosaic and sterility, (ii) mild mosaic and partial sterility, and (iii) chlorotic ring spots with no sterility (Figure 2). The virus strain also significantly influences symptom expression. For instance, genotype ICP2376 infected with SMD showed chlorotic ring spots with the Patancheru strain, whereas it exhibited severe mosaic with sterility symptoms upon infection with the Bengaluru and Coimbatore strains [12,25,29,30,31]. SMD incidence varies due to pathogen variability. In India, among ten distinct strains of *Pigeonpea sterility mosaic virus* (PPSMV) identified, strains from Bengaluru, Dholi, Vamban, and Varanasi appear to be more virulent than those originating from Badnapur, Hyderabad, Pantnagar, Kanpur, Ludhiana and Faizabad [32,33,34,35]. Additionally, SMD-infected plants aggravate powdery mildew (*Oidiopsis taurica*) infection [36] and spider mite (*Schizotetranychus cajani*) infestation [37], compounding the damage.

### 2.1. The Vector-Aceria cajani

SMD of pigeonpea is transmitted by an eriophyid mite, *Aceria cajani* Channabasavanna (Arthropoda: Acari: Arthropoda) [38] (Figure 3) but not through sap, pollen, seed, or soil [22,39,40,41,42,43]. *Aceria cajani* is microscopic and measures about 200–250 µm in length, with a short life cycle of about two weeks that includes an egg and two nymphal stages [44,45]. These mites are predominantly found on young symptomatic leaves of SMD-infected plants, residing on the undersurface of leaves and concentrating towards the petiole [28,46,47,48,49], suggesting a beneficial relationship between the mite vector and the virus [28]. The mites possess a very short stylet (~2.03 µm) and generally feed on epidermal and underlying mesophyll cells, acquiring virus only from infected cells [50]. Their direct feeding does not cause any obvious damage to pigeonpea [45]. *Aceria cajani* is highly host-specific, primarily confined to pigeonpea and a few of its wild relatives [35,45]. In nature, these mites can also be seen on *Hibiscus panduriformis* adjacent to the SMD-infected pigeonpea field [51]. 

*Aceria cajani* dispersal is passive, primarily aided by wind currents, with the mites blown up to 2 km from the source of inoculums [52]. A mean temperature of 20–30 °C is favorable for mite proliferation, while higher temperatures and heavy rains are unfavorable [53,54]. *Aceria cajani* transmits PPSMV in a semi-persistent manner, with a single mite having a transmission efficiency of up to 40%; however, more than five mites are required for 100% transmission. It requires a minimum acquisition access period (AAP) of 15 min and an inoculation access period (IAP) of 90 min for the successful transmission of the PPSMV. The mites are virulent for about 6–13 h after virus acquisition, and the virus is neither propagative nor transovarially transmitted [43,44]. Accurate identification of eriophyid mites based on morphological characters is difficult due to their small size. A diverse range of molecular markers has been used to analyze different strains (biotypes) within a species. Analysis of ribosomal DNA internal transcribed spacer (rDNA-ITS) sequence of *Aceria cajani* collected from different locations in India, Nepal, and Myanmar revealed little to no sequence divergence amongst them [48,55]. This probably indicates that there is a single biotype of *Aceria cajani* in transmitting PPSMV, and that did not differ much in their transmission.

### 2.2. SMD Transmission

Experimentally, PPSMV can be transmitted by leaf stapling method onto pigeonpea, Wild *Cajanus* species, and *Phaseolus vulgaris* cvs. Top crop, Kintoki and Bountiful, as well as *Chrozophora rottleri* thus aiding in mite transfer to test genotypes and their feeding transfers [35,43,56]. Mechanical sap inoculation to *Nicotiana benthamiana* and *N. clevalandi* was successful. However, attempts to sap transmit PPSMV onto pigeonpea were not successful [35,57]. Even purified PPSMV preparation was not infectious [58]. However, PPSMV can be experimentally transmitted to pigeonpea by grafting [23,59]. *Hibiscus panduriformins*, *Oxalis corniculata*, and *Canavis sativa* may act as refuge for mite survival in transit, thereby aiding the spread of SMD [51,53]. Though PPSMV can infect plants outside the genus *Cajanus* due to the host specificity of its mite vector to *Cajanus cajan* and a few of its wild relatives restricts its potential source of SMD inoculum under field conditions. The perennial, volunteer, and ratooned pigeonpea may provide a pool for mite vectors and viruses, thus serving as a primary source of inoculum for SMD [60].

### 2.3. SMD Etiology and Detection

The causal agent of SMD has remained elusive for several decades. Despite continuous efforts by various laboratories, the identification of the etiology has been unsuccessful, posing a significant challenge to the scientific community [27,31,61]. Based on symptoms and mode of transmission, it was suggested that the causal agent of SMD likely involves a virus or virus-like agent [21,61]. Supporting this hypothesis, several studies have demonstrated that there is no involvement of mite toxemia or fungi, bacteria, nematode, Phytoplasma, or viroid in causing SMD in pigeonpea [27,32,41,42,61].

Though SMD of pigeonpea was first identified in 1931 [11], the etiology remained a major conundrum until a breakthrough in 1999 by Kumar and his coworkers [29,30,62] leading to the identification of the causal organism provisionally named *Pigeonpea sterility mosaic virus* (PPSMV), a tenui-like virus, with highly flexuous and branched filaments virus-like particles (VLPs) measuring about 3–10 nm diameter. The purified virus preparation contained a 32 kDa major protein and up to seven RNA segments of size 6.8–1.1 kb [58,62,63]. Ultra-structural studies of thin sections of SMD-infected pigeonpea cultivars (ICP8863 and ICP2376) revealed two types of inclusions: (i) quasi-spherical membrane-bound bodies (MBBs) of 100–150 nm diameter associated with amorphous electron-dense material (EDM) and (ii) fibrous inclusions (Fis) found adjacent to the cell nucleus. In PPSMV-infected *Nicotiana benthamiana* sections, only MBBs and EDM were observed with no Fis. These structures bear a resemblance to those infected by eriophyid mite-transmitted *High plains virus* (HPV) [58,63].

Polyclonal antibodies developed in rabbits against purified PPSMV virus-like particles (VLPs) proved effective in detecting PPSMV in plants through direct antigen coating (DAC)-ELISA and double antibody sandwich (DAS)-ELISA [35,55,58,63,64]. In the ELISA test, samples from hundreds of healthy or uninoculated pigeonpea plants consistently yielded negative results, providing circumstantial evidence that PPSMV is the causal agent of the SMD of pigeonpea. Based on the above distinctive characteristics, PPSMV is presumed to form a separate genus in virus taxonomy [58,63]. Detection of PPSMV was successful in SMD-affected pigeonpea plants infected experimentally by *A. cajani,* through grafting, and naturally in the field at several different locations in India. Additionally, PPSMV was detected in infected accessions of wild pigeonpea, *Cajanus scarabaeoides*. Detection was also accomplished in *N. benthamiana* and *N. clevelandii* infected via mechanical sap inoculation [57]. Viruliferous mite extracts were found positive for PPSMV by DAS-ELISA and dot immunobinding assay (DIBA) [43,55]. Polyclonal antisera, developed to unique peptide sequences specific to nucleocapsid of PPSMV-1 (PPI_5–20_ MPSKTPFSNMPAASKK^*^) and PPSMV-2 (PPII_299–314_ *STFLPALEADRLASLP) (peptide conjugated to KLH at cysteine residue, i.e., “*” position) were utilized to detect PPSMV-1 and PPSMV-2 [25]. Next-generation sequencing using Illumina technology of double-stranded (ds)-RNAs recovered from pigeonpea leaves infected with SMD revealed that the PPSMV genome is segmented and is comprised of five encoding five peptides. These peptides show homology to the polypeptides encoded by the corresponding RNA segments of the *Emaravirus* genus and known members of the Bunyaviridae family, demonstrating a consistent relationship with all emaraviruses and, in particular, *Fig mosaic virus* (FMV) and *Rose rosette virus* (RRV) [65].

The earlier observation of 5-7 RNA segments in the PPSMV genome [58] was inconclusive as it was based on analysis of dsRNA, this was likely compromised by the presence of mixed infection with other PPSMV strain or another virus. During a survey of pigeonpea fields in Hyderabad, some plants showing typical SMD symptoms tested negative for PPSMV by RT-PCR [65,66]. Further, testing of SMD-infected samples collected from the Chevella region near Hyderabad revealed the presence of more than six dsRNA segments. This finding suggested the potential association of yet another mite-transmitted virus with SMD in pigeonpea [25]. 

Deep sequencing of dsRNAs recovered from pigeonpea leaves affected with mosaic disease revealed the presence of another Emaravirus, *Pigeonpea sterility mosaic virus-2* (PPSMV-2) (PPSMV will be referred to as PPSMV-1 hereafter), a six-segmented RNA virus. Phylogenic analysis of nucleotide and amino acid sequence of PPSMV-2 showed the highest identity with FMV and RRV and distinct from PPSMV-1. The separate branching of PPSMV-1 and PPSMV-2 in the phylogenic tree suggests that these two emaraviruses infecting pigeonpea have followed independent evolutionary paths. Based on the molecular and morphological features, both viruses were placed in the genus *Emaravirus* within the family Fimoviridae of the genus *Emaravirus* [25,66,67].

### 2.4. PPSMV Genome Organization and Genetic Relationship

Oligonucleotide primers, SM-1 (5′-ACATAGTTCAATCCTTGAGTGCG-3′) and SM-2 (5′-ATATTTTAATACACTGATAGGA-3′) derived from the nucleotide sequence of RNA-5 segment, specifically amplified a 321 bp product from purified PPSMV RNA preparation and total RNA extracts of PPSMV-infected pigeonpea leaves and *N. benthamiana* plants by reverse transcription-polymerase chain reaction (RT-PCR) [63,68]. However, subsequent investigations revealed that the above primers (SM-1 and SM-2) were amplifying a partial sequence of the RNA-3 segment of PPSMV [25,67].

The novel *Emaravirus* species, PPSMV associated with SMD of pigeonpea composed of five large RNA fragments. RNA-1 (7022 nt) encodes for RNA-dependent RNA polymerase (RdRP, p1), exhibiting 37 to 54%amino acid similarity with the RdRp of other emaraviruses. RNA-2 (2223 nt) encodes for a glycoprotein precursor (GP, p2) and shares 31 to 45% identity at the amino acid level with GPs of *Fig mosaic virus* (FMV), *Rose rosette virus* (RRV), *Raspberry leaf blotch virus* (RLBV) and *European mountain ash ringspot-associated virus* (EMARaV). RNA-3 (1442 nt) encodes for the nucleocapsid protein (NCP, p3), showing 25 to 44% amino acid identity with NCP proteins of RRV, FMV, EMARaV, RLBV, and *Maize red stripe virus* (MRSV). RNA-4 (1563 nt, p4) encodes the movement protein (MP, p4), sharing 41% amino acid similarity with the MP encoded by RRV. RNA-5 (1801 nt) encodes a polypeptide (p5) of unknown function, which shares 33% of sequence identity with the RNA-5 encoded protein of FMV (Elbeaino et al., 2014). A phylogenetic tree, constructed with the complete amino acid sequences encoded by the *PPSMV* and the homologous proteins encoded by the RNA segments of members of the Bunyaviridae family, showed a consistent relationship of PPSMV with all emaraviruses, particularly with FMV and RRV [25,65,67].

Subsequently, another eriophyid mite transmitted *Emaravirus* has been characterized from pigeonpea leaves associated with SMD and provisionally named PPSMV-2, which contains six RNA fragments. Like PPSMV-1, corresponding RNA particle codes for similar polypeptides [66]. The two polypeptides (p5 and p6) with unknown (UK) functions could be involved in the virus life cycle through different roles, including the transmission by eriophyid mites [69]. The phylogenetic trees developed showed that emaraviruses clustered in two discrete clades: one containing WmoV and RLBV, and the second comprised of all other emaraviruses. PPSMV-1 and PPSMV-2 originate from a common ancestor branched into different clades, distinct from each other and diversifying as two distinct viruses infecting a common host. Closely related PPSMV-2 and FMV are present in the same clade, reflecting an evolutionary relationship, while PPSMV-1 and RRV formed as closely associated taxa in a separate clade with a common ancestor from other emaraviruses [8,25,66].

There was speculation regarding the number of RNA segments in the PPSMV-1. Hits recognized in deep sequencing analysis, corresponding to RNA-5 matched to PPSMV-2. This was confirmed by RT-PCR analysis using RNA-5-specific primers, suggesting that PPSMV-1 may not possess RNA-5 as a genomic segment. In further investigations, some pigeonpea plants of cv. ICP 8863, when inoculated with PPSMV-P sub-isolate Chevella, showed the presence of PPSMV-1 alone without PPSMV-2. In these plants, RNA-5 was not amplified when specific primers were used, confirming the absence of this segment. Phylogenetic analysis revealed that RNA-5 of PPSMVs formed a single clade with a common ancestor, suggesting that earlier reports of PPSMV-1 containing five genomic RNAs may not be convincing (Figure 4) [25]. The variability in emaraviruses associated with SMD in India indicated the independent existence of PPSMV-1 and PPSMV-2 alongside mixed infections. Phylogenetic analysis of nucleotide of RNA-3 of PPSMV isolates showed significant sequence variability (Figure 5) [8,67,70].

## 3. Epidemiology of SMD

The epidemiology of SMD is complex and involves the virus strain mite vector, pigeonpea cultivar, and the unpredictable environmental conditions of the semi-arid tropics. In India, SMD occurs every year in almost all pigeonpea-growing regions, but the disease incidence varies widely across different regions and seasons [27]. Conflicting reports exist regarding the influence of climatic conditions on SMD epidemiology [47,53]. Crops grown under irrigated conditions or in proximity to irrigated fields are particularly vulnerable to early SMD infection [71]. 

SMD is not seed or soil-borne; it is solely introduced by its mite vector. Diseased plants left in the field after harvest, on the field bank, in the kitchen garden, or wild relatives of pigeonpea such as *C. platycarpus* and *C. scarabaeoides* serve as reservoirs for the mite vector and the virus during the offseason, thus acting as an inoculum for SMD [5,35,60]. In rainfed pigeonpea cultivation, the primary source of inoculum is the stubbles left in the field after harvesting, volunteer plants, or plants near water sources or in the shade as these plants maintain foliage and harbor both the vector mite as well as the virus. After the early rains, these plants produce a new flush, thus creating favorable conditions for multiple cycles of the mite vector before spreading to newly planted crops. The spread of the disease within the field depends on pigeonpea cultivar, plant age, climatic factors, and mite population. In regions where one pigeonpea crop is followed by a wide time gap, and volunteer crop is not common, the reappearance of the SMD is unknown. However, in such regions, it is suspected that the wild relatives of pigeonpea, such as *C. platycarpus* and *C. scarabaeoides*, serve as the primary source of inoculum, and viruliferous mites assisted by wind currents aid in the spread of the disease [5,6]. Weed species such as *Hibiscus panduriformins*, *Oxalis corniculata*, and *Canavis sativa* may act as refuges for mite survival in transit and may, therefore, contribute to the spread of SMD [51,53]. 

A weather-based forecasting study of SMD was conducted for SK Nagar (Gujarat), Gulbarga/Kalburgi (Karnataka), Rahuri (Maharashtra) and Vamban (Tamil Nadu). The Hybrid models, specifically Autoregressive Integrated Moving Average (ARIMA)- Support Vector Regression (SVR) and ARIMA-Artificial Neural Network (ANN), outperformed individual models in predicting disease incidence at SK Nagar, Gulbarga/Kalburgi, and Vamban. However, at Rahuri, individual models demonstrated better compared to the hybrid model with ARIMA. The use of the ARIMA-SVR hybrid model was found to be particularly applicable under conditions where the seasonal mean severity of SMD exceeds 1%. Conversely, the SVR model can effectively predict disease incidence when it is less than 1% [72].

## 4. Screening Techniques for SMD Resistance

The PPSMV is not mechanically sap transmissible onto pigeonpea [6,35,57]. Screening for SMD in pigeonpea genotypes is commonly performed using two methods: (i) leaf stapling [73,74] and (ii) infector hedge/row [75,76] facilitating PPSMV transmission through the mite vector. Vector mite transmission of PPSMV occurs if the genotype is susceptible to both the virus and its vector. Failure of virus transmission suggests that the tested genotype could possess resistance to either the vector, the virus, or both. Genotypes resistant to mite inoculation are further assessed for virus resistance using the “grafting technique” [23,59]. PPSMV exhibits four different host responses in pigeonpea germplasm: (a) severe mosaic and sterility (ICP8863), (b) mild mosaic and partial sterility (ICP8862), (c) chlorotic ringspot (ICP2376 to Patancheru isolate), and (d) no visible symptoms (ICP7035) [6]. Based on SMD incidence, test genotypes are categorized into four groups: (i) resistant (R) (<10% SMD incidence), (ii) moderately resistant (MR) (10.1–20% SMD incidence), (iii) susceptible (S) (20.1–40% SMD incidence), and highly susceptible (HS) (>40.1% SMD incidence) [10]. The response of different pigeonpea genotypes against SMD was assessed and presented in Table 1 [77].

### 4.1. Leaf Stapling Method

The leaf stapling technique developed by Nene and Reddy [73,74] allows the rapid screening of pigeonpea genotypes under both field and glasshouse conditions. Young SMD-infected leaflets from susceptible genotype (e.g., ICP8863) collected in a moist cloth bag were observed for mite infestation to ensure a minimum of 10 mites per leaf. The mite-infested leaflets were then stapled onto test plants at the two- to three-leaf stage in such a way that the undersurface of the diseased leaflet comes in contact with both surfaces of the test plant by folding, anchoring mites for transfer. The feeding of these mites results in PPSMV transmission onto the test plant (Figure 6). Test genotypes are subsequently observed for initial disease symptom development, and the observations were taken 60 days after planting.

### 4.2. Infector Hedge/Row Method

This field screening method is designated for large-scale pigeonpea genotype screening through natural infestation. Four to five rows of a susceptible genotype (e.g., ICP8863) are planted across the wind direction and upwind one side of the field one month prior to the actual trial begins (Figure 7a). SMD is maintained in these rows using the leaf stapling method. Subsequently, test genotypes and a susceptible genotype (used as an indicator/infector row to monitor disease spread) are planted in the field at a 10:1 ratio (Figure 7b). The viruliferous mites from the infector hedge are carried by the wind, facilitating disease transmission. Successful screening is determined when infector rows between test plants reach 100% infection. Disease incidence is recorded twice: first in the seedling stage (30–40 days after planting) and then at maturity. SMD inoculum can be maintained on pruned foliage of the infector hedge for the next crop season [75,76]. 

### 4.3. Petiole-Grafting

Genotypes resistant to mite inoculation were evaluated for virus resistance using the petiole-grafting technique [23,59]. Fourteen to sixteen-day-old plants grown in growth chambers were used for grafting. The process involved the use of an infected leaflet from the PPSMV-infected ICP8863 genotype as a scion. These leaflets were freed from mites through miticide spraying. The terminal end of the test genotype was excised, and a 5–10 mm incision down the center of the stem was made using a scalpel blade. The scion was trimmed into a wedge shape and inserted into the stem slit of the stock plant, tightly bound with cellophane tape/grafting clips, and covered with polythene bags to maintain humidity for seven days inside a mite-proof growth cabinet. The test genotypes were examined for the development of disease symptoms and assayed for PPSMV by DAS-ELISA or RT-PCR [62] at 14, 20, and 35 days after grafting. 

## 5. SMD Management

Various strategies have been explored to mitigate SMD incidence, including the application of pesticides to delay the onset of infection and disease spread, as well as control through cropping methods and host-plant resistance. In field conditions, SMD spreads to pigeonpea exclusively through eriophyid mites, the interference of vector transmission through the application of pesticides such as chinomethionate, dinocap, dicofol, fenazaquin, golecron, sulfur, metasystox, monocrotophos, morestan, profenophos, proporgite or wettable sulfur was proven effective in reducing SMD [5,12,41,60,78,79,80,81,82,83,84,85,86]. Seed treatment with 25% carbofuran or soil application was shown to protect the crop from SMD infection for up to 75 days. Water spray also demonstrated a 70% reduction in the vector mite infestation [41,52]. Proper detection and monitoring of vector mites are crucial for effective control because, usually, the presence of *A. cajani* is determined based on SMD symptoms induced by PPSMV. 

Though chemical management of SMD is effective, it is an expensive option for subsistence farmers considering pigeonpea cultivation with low input cost. Studies on different sowing dates, plant density, spacing, intercropping, or border crops have shown no significant effect on SMD incidence [18,53,81,87,88]. Destruction of sources of SMD inoculum prior to the cropping season can reduce SMD incidence and/or delay the early onset of the disease, thereby minimizing its impact. However, such practices are seldom followed due to the farmers’ preoccupation with other revenue-generating activities, lack of resources, and labor shortages in marginal farming conditions where the crop is predominantly cultivated. 

Management of SMD through host-plant resistance (HPR) has been prioritized, as it requires no special expertise from growers. HPR is the most reliable and cost-effective approach for SMD management in pigeonpea. The identification of sources of resistance to SMD began with the pigeonpea landrace Sabour 2E in India [61,81]. Subsequent efforts for identifying sources of resistance were initiated at the International Crops Research Institute for the Semi-Arid Tropics (ICRISAT) in collaboration with National Agricultural Research Systems (NARS). From the global pigeonpea collection at ICRISAT, consisting of 13,015 pigeonpea accessions, 326 resistant and 97 tolerant lines were identified [61]. The selected 326 resistant lines were evaluated at various locations in India; only 10 genotypes exhibited resistance across all locations. The variation in the resistance levels could be attributed to the occurrence of different virus strains. In India, 10 distinct strains of PPSMV have been identified, with strains from Bengaluru, Dholi, Vamban and Varanasi appearing to be more virulent than those at Badnapur, Hyderabad, Pantnagar, Kanpur, Ludhiana and Faizabad [32,33,34,35]. The dynamic nature of the PPSMV has guaranteed the use of strain-specific sources of resistance in crop improvement. Thus, there is a pressing need to identify strain-specific resistance sources for more effective management strategies. 

During a five-year period from 1978 to 1983, 88 pigeonpea germplasm lines that had shown resistance at Patancheru, Badnapur, and Kanpur isolates were further tested with isolates from 10 different locations in India (Badnapur, Bengaluru, Dholi, Faizabab, Kanpur, Ludhiana, Pantnagar, Vamban and Varanasi) through joint ICRISAT-Indian Council of Agricultural Research (ICAR) uniform trails for broad-based resistance to pigeonpea sterility mosaic. Among the test lines, ICP 7867, 10976, and 10977 exhibited resistance across all 10 isolates, while ICP 11146 showed resistance or tolerance against nine isolates except the one derived from Dholi, and ICP 10983 showed resistance or tolerance against eight isolates, except Bengaluru and Varanasi [89]. Novel resistance sources in pigeonpea genotypes and released varieties are presented in Table 2 and Table 3.

Adequate levels of SMD resistance are scarce within the cultivated pigeonpea gene pool. However, wild relatives of pigeonpea have been shown to possess a high level of resistance to various biotic constraints [106]. The global pigeonpea germplasm collection at ICRISAT comprises over 270 accessions of 47 wild species related to the genus *Cajanus*. Among the 115 accessions representing six wild *Cajanus* species, *C. albicans*, *C. cajanifolius*, *C. lineatus*, *C. platycarpus*, and *C. scarabaeoides* were screened for resistance to PPSMV isolates in Patancheru, Bengaluru, and Coimbatore. Fifteen accessions, ICP 15164, 15615, 15626, 15684, 15688, 15700, 15701, 15725, 15734, 15736, 15737, 15740, 15924, 15925 and 15926 were identified as posing broad-based resistance. Some of these accessions also showed resistance to pest damage, cyst nematode, and wilt. Except for *C. platycarpus*, the tested species belong to the secondary gene pool and are inter-fertile by traditional breeding. Therefore, the resistance found in these accessions is transferable to pigeonpea through a conventional breeding approach. The resistance in few wild *Cajanus* species accessions such as *C. cajanifolius* (IC272730, 552553, 552557, 552601), *C. platycarpus* (IC525192, 550320, 550321), *C. scarabaeoides* (IC308694, 308802, 325999, 382113), *C. albicans*, *C. volubilis*, *C. sericeus*, and *C. lineatus* was reconfirmed [97].

## 6. Application of Molecular Markers

Sequencing-based bulk segregant analysis (Seq-BSA) was used to map resistant genes for Fusarium wilt (FW) and SMD in pigeonpea. Resistance (R) and susceptible (S) from extreme recombinant inbred lines of ICPL 20096 × ICPL 332 were sequenced. Seq-BSA has provided seven candidate SNPs for FW and SMD resistance in pigeonpea, which are useful for genomics-assisted breeding in pigeonpea [107]. A diagnostic marker kit was developed with 10 markers for the identification of SMD-resistant lines [108]. A 50 hypervariable pigeonpea-specific simple sequence repeat (AHSSR) markers were screened to identify genomic regions associated with resistance to SMD through bulk segregant analysis (BSA) approach in 84 RILs derived from a cross ICP 8863 (S) × BRG 3 (R). Three SSR markers, such as AHSSR 50_150_, AHSSR 34_130_, and AHSSR 20_125_, differentiated resistant bulks, susceptible bulks and RIL individuals used for constituting bulks. Single marker analysis (SMA) showed that these three markers were closely associated with SMD resistance [109].

Bulk segregant analysis (BSA) was employed to identify the RAPD marker linked to SMD and to develop a SCAR for the SMD resistance gene of ICPL 7035. Of the 200 RAPD primers screened, OPA1880 showed polymorphism in resistant and susceptible lines of a cross ICPL 7035 × ICPL 8863, indicating that OPA1880 was associated with SMD resistance in ICPL 7035. A SCAR marker, SCAR 816 (16 f/r), developed from end sequences, was present in all generations (parents, F1 and F2) and is found useful to identify the transferred SMD resistance gene to a line [110]. To identify the gene(s) or QTLs linked with SMD resistance, two F_2:3_ populations such as ICP 8863 × ICPL 20097 (segregating for Patancheru SMD isolate) and TTB 7 × ICP 7035 (segregating for Patancheru and Bengaluru SMD isolates) were phenotyped to respective SMD isolates. More than 3000 SSR markers on parental genotypes of each mapping population, intra-specific genetic maps comprising 11 linkage groups and 120 and 78 SSR loci were developed for ICP 8863 × ICPL 20097 and TTB 7 × ICP 7035 populations, respectively. Composite interval mapping (CIM) based QTL analysis by using genetic mapping and phenotypic data provided four QTLs for the Patancheru isolate, and two for the Bengaluru isolate. This identification of different QTLs for resistance to Patancheru and Bengaluru isolates indicates the involvement of different genes conferring the resistance [111]. From the three mapping populations, including two recombinant inbred lines PRIL_B (ICPL 20096 × ICPL 332), PRIL_C (ICPL 20097 × ICP 8863), and one F2 (ICP 8863 × ICPL 87119), a total of 212,464 SNPs were identified in PRIL_B, 89,699 in PRIL_C, and 64,798 in the F2 population through genotyping-by-sequencing approach. Composite interval mapping-based QTL analysis identified three major QTLs across the three populations. Particularly, one candidate genomic region identified on CcLG11 seems to be a promising QTL for molecular breeding in developing superior lines with enhanced resistance to SMD [112]. 

## 7. Future Line of Work

The management of SMD of pigeonpea presents a complex challenge. With the recent discovery of a second *Emaravirus* associated with SMD, the identification and development of pigeonpea varieties resistant to SMD has become a critical focus for future research. The complex nature of the disease is compounded by the diverse virus strains found across India, prompting the need for the development of strain-specific resistant sources to enhance the adaptability of pigeonpea cultivars to varying disease pressures. Reevaluating mini-core and wild *Cajanus* collections becomes vital to identifying new sources of resistance, considering the evolving dynamics of the virus strains over time. Recognizing the socio-economic significance of pigeonpea, there is a pressing need to channel efforts into developing new high-yielding lines. This initiative aligns with the broader goal of increasing agricultural productivity to support the well-being of subsistence farmers who heavily depend on pigeonpea cultivation. In addition, with the challenges posed by SMD, the simultaneous occurrence of Fusarium wilt (FW) underscores the importance of integrated resistance strategies. Future efforts should prioritize the development of pigeonpea varieties that exhibit robust resistance to both SMD and FW, offering a holistic approach to sustainable disease management. Adopting new breeding approaches, such as speed breeding, accelerates and enables quicker development of lines, which enhances the efficiency of developing SMD-resistant lines, aligning with the urgency of addressing disease challenges in pigeonpea cultivation. Moreover, the application of molecular markers in breeding programs is instrumental in developing resistant lines. Markers such as SNP, SSR, and SCAR facilitate the precise identification and selection of pigeonpea genotypes with enhanced resistance traits, streamlining the breeding process. This targeted approach accelerates the development of SMD-resistant pigeonpea varieties, ensuring the efficient deployment of genetic resources and contributing to sustainable crop improvement programs. The Clustered Regularly Interspaced Short Palindromic Repeats (CRISPR)-Associated (Cas) system provides precise gene editing capabilities and has successfully been employed in managing several plant diseases [113]. This approach offers a complementary strategy to traditional breeding methods, potentially enabling the rapid development of resistant pigeonpea lines to specific virus strains, thus contributing to sustainable disease management in pigeonpea cultivation.

Climate change directly influences the distribution, severity, and emergence of plant diseases. Future research on climate change and its impact on pigeonpea SMD resistance should focus on understanding the ecology and behavior of SMD vectors. Additionally, efforts should be directed toward integrating climate data with genetic information to develop pigeonpea varieties resistant to SMD. Implementing integrated pest management (IPM), strategies adopted to mitigate SMD impact in varying climatic scenarios is essential. Collaborative interdisciplinary research and knowledge sharing among scientists, breeders, and agricultural extension services will be vital for developing sustainable solutions aimed at combating SMD. 

## Figures and Tables

**Figure 1 plants-13-02146-f001:**
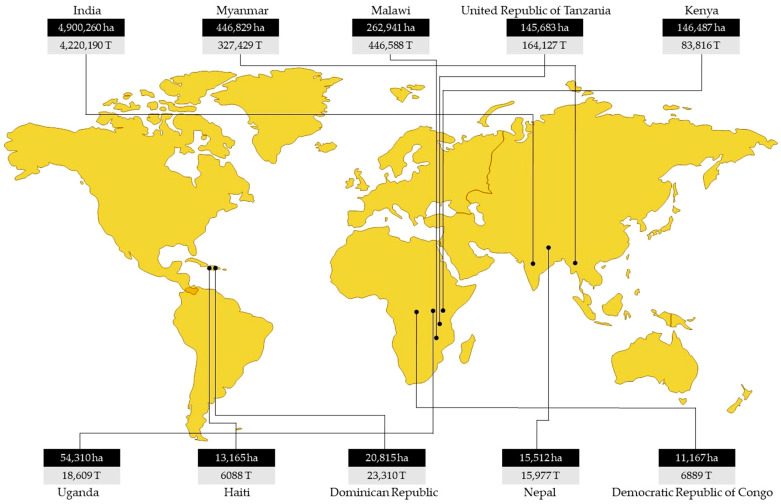
Area (hectares (ha)) and production (tonnes (T)) of major pigeonpea producing countries [2].

**Figure 2 plants-13-02146-f002:**
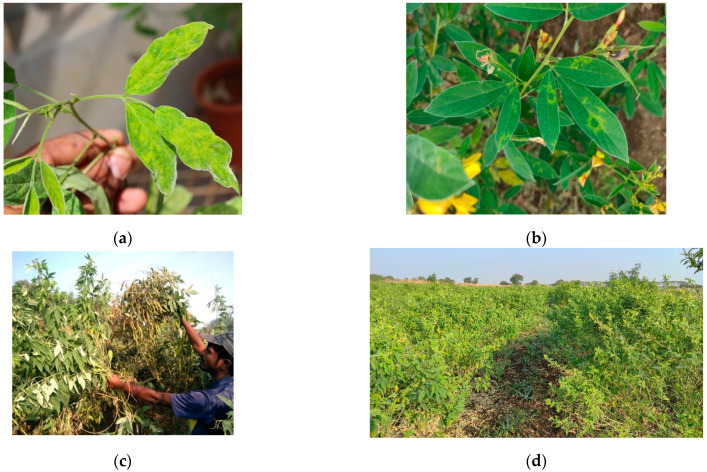
Pigeonpea sterility mosaic disease symptoms (**a**) mosaic; (**b**) chlorotic ringspots; (**c**) partial sterility; (**d**) complete sterile pigeonpea field.

**Figure 3 plants-13-02146-f003:**
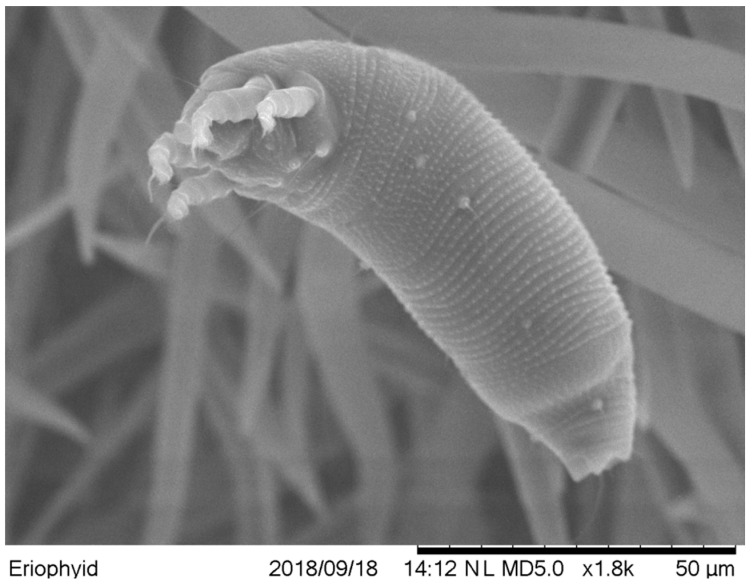
Scanning electron microscopic (SEM) picture of *Aceria cajani*, mite vector of pigeonpea sterility mosaic disease.

**Figure 4 plants-13-02146-f004:**
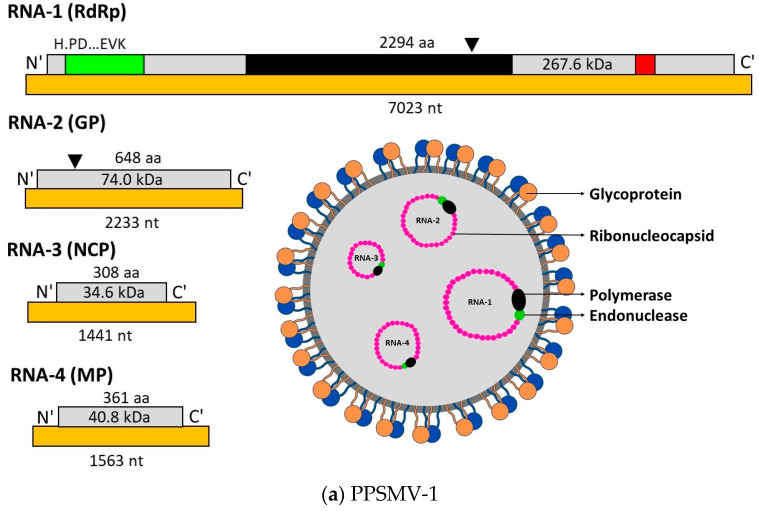
Schematic representation of PPSMV-1 (**a**) and PPSMV-2 (**b**) virus structures and RNA particles. PPSMV-1 is composed of 4 RNA segments, and PPSMV-2 is composed of six RNA segments. RNA-1 encodes for polymerase protein (RdRp), RNA-2 encodes for glycoprotein (GP) cleavage site denoted by black triangles, RNA-3 encodes for nucleocapsid protein (NCP), and RNA-4 encodes for movement protein (MP). RNA-5 and RNA-6 of PPSMV-2 encodes for proteins with unknown (UK) function. RdRp of both the viruses are similar, showing the location of endonuclease (green), Bunya RdRp (black), and proposed cap-binding site (red) subunits (pictures are reproduced [25]).

**Figure 5 plants-13-02146-f005:**
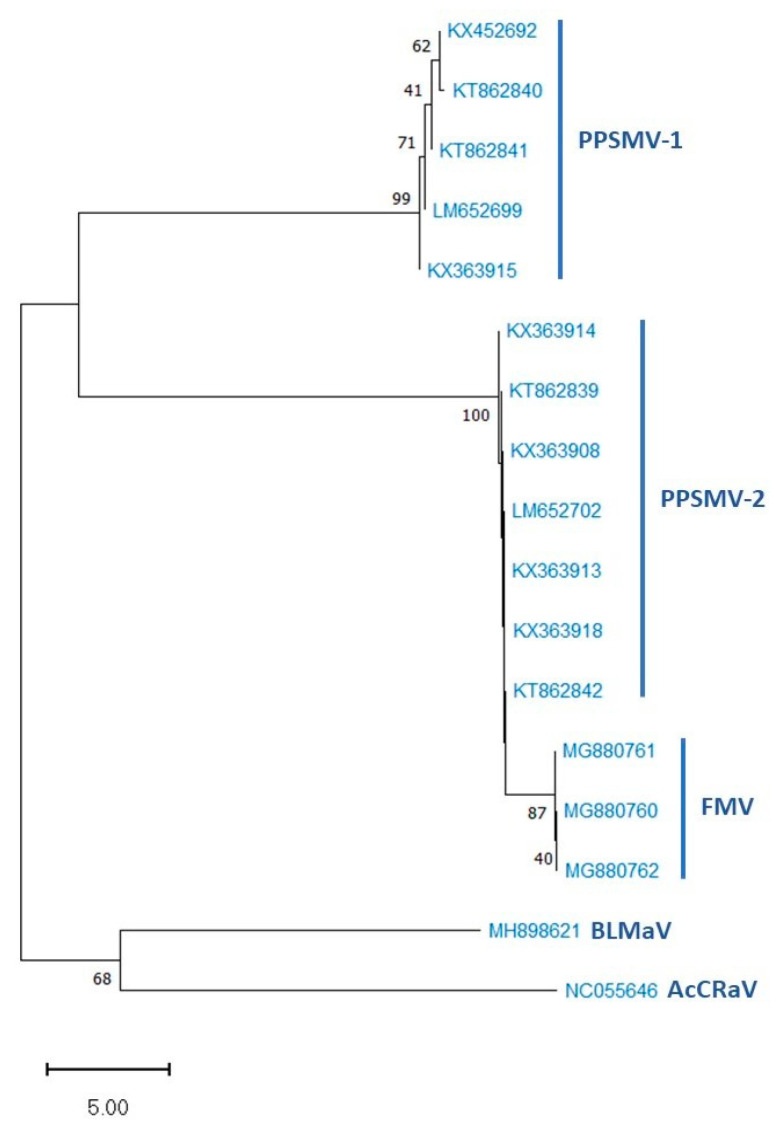
Phylogenetic tree of nucleotide sequences of RNA-3 segments of PPSMV-1, PPSMV-2, and related *Emaravirus* species with bootstrap values based on 1000 replicates. The scale bar represents 5.00 substitutions per nucleotide position.

**Figure 6 plants-13-02146-f006:**
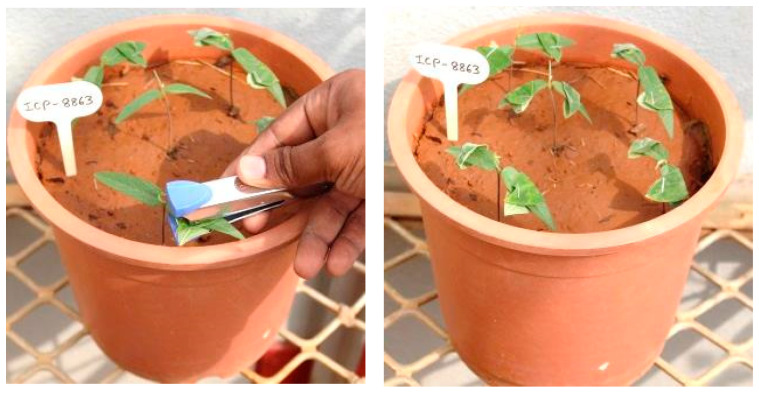
Leaf-stapling method of screening for sterility mosaic disease resistance.

**Figure 7 plants-13-02146-f007:**
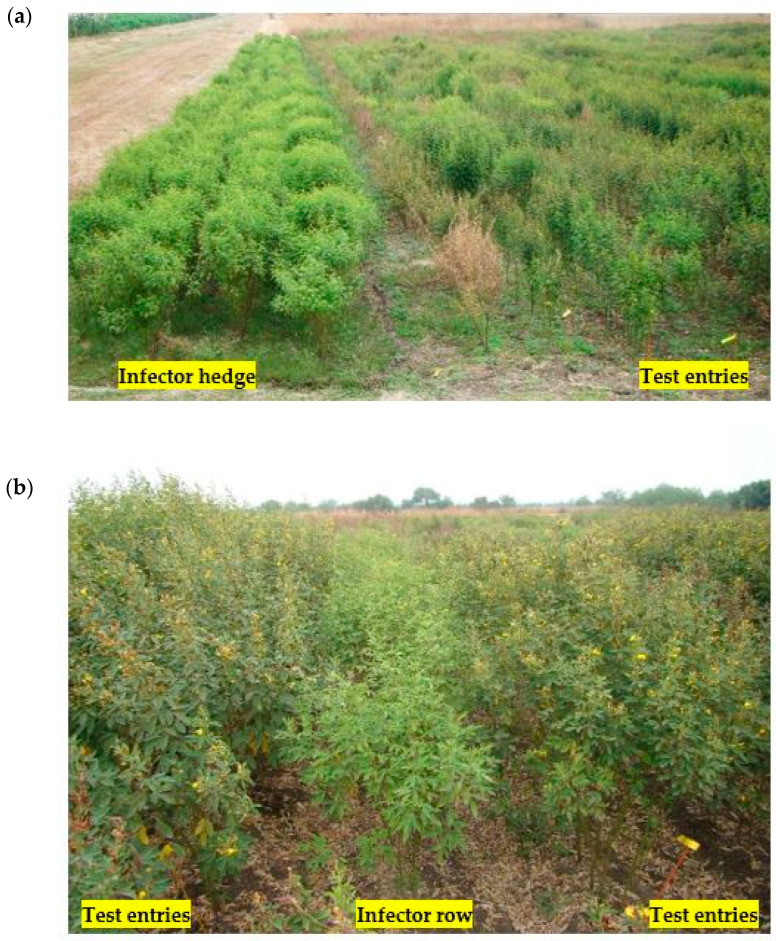
(**a**) Infector-hedge method (4–5 rows) of screening for sterility mosaic disease resistance. (**b**). Infector-row method (10:1) of screening for sterility mosaic disease resistance.

**Table 1 plants-13-02146-t001:** The response of pigeonpea host-differentials to *Pigeonpea sterility mosaic virus* (PPSMV) isolates in India.

Sl. No.	Variety	Patancheru	Dholi	Varanasi	Badnapur	Rahuri	Bengaluru	Coimbatore	Dharwad	Tirupati	Vijayapura	Warangal
1	Bahar	R	R	R	R	R	R	R	R	HS	R	R
2	BRG 1	R	R	R	R	R	R	MR	R	R	MR	R
3	BRG 2	R	R	R	R	R	S	MR	R	MR	MR	R
4	BRG 3	R	R	R	R	R	R	R	R	R	MR	R
5	BRG 5	*	R	R	R	MR	S	R	MR	R	MR	R
6	BSMR 736	R	R	MR	R	R	R	MR	R	R	R	R
7	ICP 7035	R	MR	R	R	R	R	R	R	HS	R	S
8	IPA 8F	R	R	R	R	HS	R	R	R	R	R	R
9	Purple 1	MR	R	HS	R	R	S	R	MR	HS	R	S
10	ICP 2376	R	R	R	R	R	MR	R	S	HS	R	S
11	ICP 8863	HS	HS	HS	HS	HS	HS	S	HS	HS	MR	HS

R—resistant (<10% SMD incidence); MR—moderately resistant (10.1–20% SMD incidence); S—susceptible (20.1–40% SMD incidence); and HS—highly susceptible (>40.1% SMD incidence) [10]. * not tested.

**Table 2 plants-13-02146-t002:** Available pigeonpea-resistant sources against sterility mosaic disease (SMD) in India.

Sl. No.	Genotypes/Cultivars	Reference
1.	NP (WR) 15, P-435, P-1100, P-1289, P-1778 and P-2621	[81]
2.	L-3 and P-4875	[90]
3.	ICRISAT-3783, 6986, 1137, 2719, 7119, HY-3c, ICP-7035 and *Atylosia lineata*	[74]
4.	ICRISAT-3784, 5449, 6497, 7035, 7119, Pant 8-76, 8-77, E-41, P-4785 and L-26	[91]
5.	ICP-7035, 3782, 6986, 6997, 7119, 7197, 7867, 7942 and 8136	[28]
6.	ICP 7378, 2S2	[46]
7.	Bahar, ICPLC-88046, DA-35, K32-1, Pusa-14, 19, Gant-9005, DA-I 1, 32,33	[92]
8.	ICP 7035	[93]
9.	ICPL 787119	[94]
10.	ICP 7035 and HY3C	[95]
11.	ICP 7035, MAL 14, and MAL 19	[96]
12.	ICP’s 7869, 9045, 11015, 11230, 11281, 11910, and 14976. Combined resistance to FW and SMD- ICP 6739, 8860, 11015, 13304, 14638, and 14819	[9]
13.	BRG3, IPA8F, IPA 15-F, GT 101, ICP 7035, ICPL 87091 and JKM 189	[60]
14.	ICPL 85010, IC245198, IC45768, IC73313, IC73336, IC73342, IC73340, IC73347, IC73727, IC73731, IC73735, IC73739, IC73841, IC73879, IC73880, IC73914, IC73925, IC73747, IC245198, IC 45768, IC73332, IC74126, IC 74084, IC74107, IC74123, Sehore, DPPA84-61-3, DPPA 84-8-3, ICP 786 (IC 306500), ICP 4395, ICP6997, ICP10976, CP10977, ICP 7035 (IC 306496), ICP 8862, MA 97, Rampur, Amar, Bahar, Bageshwari, Pant A3, PantA 104, Pant A 8505, Pant A 8508, Narendra arhar 1, comp 1-ESR6, Hy 3C, Maruti, BSMR1, BSMR 2, BSMR604, BSMR736, Bhawanisagar 1, NPRR-1, ICP6997, MA 3, LRG 36, S1-3, ICPL88034 (IC245153), DPPA85-3(IC527890), DPPA85-14(IC527891), ICPL7197, ICPL7264, PUSA14, USA15, PUSA17, PUSA18, Purple-1, PR 5149, PI 397430, KA-32-1, ICPL342, ICPL88034, DPPA85-2, DPPA85-15, BWR159, MA3, MA6, KPL 44, AWR1, KAWR2, KAWR 7, KAWR 9, KAWR 73,	[97]
15.	ICP772, ICP939, ICP995, ICP1071, ICP1126, ICP1156, ICP1273, ICP1279, ICP2577, ICP2698, ICP2746, ICP3451, ICP3576, ICP4029, ICP4167, ICP4307, ICP7426, ICP4575, ICP4715, ICP6128, ICP6370, ICP6668, ICP6739, ICP6845, ICP6859, ICP6929, ICP7076, ICP7148, ICP7223, ICP6049, ICP7803, ICP7869, ICP8012, ICP8255, ICP8266, ICP8602, ICP8793, ICP8860, ICP8949, ICP9045, ICP9336, ICP9655, ICP10094, ICP10228, ICP11321, ICP10447, ICP11015, ICP11059, ICP11230, ICP11281, ICP11627, ICP11823, ICP11910, ICP12105, ICP12123, ICP12410, ICP12654, ICP13011, ICP13139, ICP13191, ICP13244, ICP13270, ICP13304, ICP13359, ICP13431, ICP13577, ICP13579, ICP13633, ICP13662, ICP13884, ICP14116, ICP14120, ICP14147, ICP14155, ICP14545, ICP14569, ICP14638, ICP14701, ICP14722, ICP14801, ICP14819, ICP14832, ICP14903, ICP14976, ICP15049, ICP15068, ICP15107, ICP15161, ICP15185, ICP12142, ICP16264 and ICP163	[98]
16.	ICPL 20094, ICPL 20106, ICPL 20098 and ICPL 20115	[10]
17.	ICPL-87119, ICPL-2376, BDN-2, PT-4-307, CORG-9701, BSMR-736, GRG-811 and BSMR-853	[99]
18.	DPP 2-89, DPP 3-182, IC 22557 and ICP 3666	[100]
19.	TDRG 59 (ICPL 99050) and Bheema	[101]
20.	BWR 153 and CRG 16-07	[102]
21.	ICPL-16086 and ICPL-16087	[35]
22.	NAM 2082, NAM 2088, NAM 2089, NAM 2162, GRG 11, TS-3R-58-53-2, GRG 152, KRG 33, BSMR 736, AND ICPL 87119	[103]

**Table 3 plants-13-02146-t003:** State-wise recommended pigeonpea varieties for commercial cultivation in India [104,105].

State	Varieties
Andhra Pradesh	Laxmi, LRG-41, LRG-38, WRG-27, WRG-53, Bahar, Pusa-9, NDA 1, WRG 65, Surya (MRG 1004)
Bihar	MA-6, Ajad, DA-11, IPA-203, Bahar, Pusa-9, Narendra Arhar-2
Madhya Pradesh	JKM-189, TJT-501, JKM-7, TT-401, BSMR-175, ICPL-87119, BSMR-736
Chhattisgarh	Rajiv Lochan, MA-3, ICPL-87119, Vipula, BSMR-853
Gujarat	GT-100, GT-101, Banas, BDN-2, BSMR-853, AGT 2
Haryana	Paras, Pusa-992, UPAS-120, AL-201, Manak, Pusa-855, PAU-881
Karnataka	Vamban-3, CORG-9701, ICPL-84031, BRG-2, Maruti (ICP-8863), WRP-1, Asha (ICPL 87119), TS-3, KM 7
Maharashtra	BDN-711, BSMR-736, AKT-8811, PKV Tara, Vipula, BDN-708, Asha, BSMR 175, Vaishali (BSMR 853)
Punjab	AL-201, PAU-881, Pusa-992, Upas-120
Uttar Pradesh	Pradesh Bahar, NDA-1, NDA-2, Amar, MA-6, MAL-13, IPA-203, UPAS 120
Rajasthan	UPAS-120, PA-291, Pusa-992, Asha (ICPL-87119), VLA -1
Tamil Nadu	Co-6, CORG-9701, Vamban-3, ICPL-151, Vamban 1, Vamban 2
Jharkhand	Bahar, Asha, MA-3
Uttarakhand	VLA-1, PA-291, UPAS 120

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
