# Peer review of "Sterility Mosaic Disease of Pigeonpea (*Cajanus cajan* (L.) Huth): Current Status, Disease Management Strategies, and Future Prospects"

_plants, 2024, doi:10.3390/plants13152146_

Round 1
Reviewer 1 Report
Comments and Suggestions for Authors
The paper represent an important findings. I only have minor suggestions on how to improve the paper
1) Is there usually a pattern of disease spread within the field?
2) How severe is the disease in terms of yield losses or impact on plant health?
3) Is there a need to confirm the presence of the pigeonpea sterility mosaic virus (PPSMV) using multiple diagnostic tests?
4) What are the common practices related to crop rotation, irrigation, and pesticide use?
5) Are there any other factors that could exacerbate SMD symptoms?
Author Response
Comment1: Is there usually a pattern of disease spread within the field?
Response 1: Disease spread in the field primarily depends on the vector mite, which is typically transferred from plant to plant by wind. There is no specific pattern of disease spread observed in the field. However, apparently initial disease symptoms often appear first in the border rows of the field, influenced by where the mite vector initially lands.
Comment 2: How severe is the disease in terms of yield losses or impact on plant health?
Response 2: Disease severity varies depending on the genotype resistance level and the stage of infection. Infections during the early stages (<45 days old plants), especially in highly susceptible varieties like ICP-8863 (Maruti), can result in complete yield loss of the crop.
Comment 3: Is there a need to confirm the presence of the pigeonpea sterility mosaic virus (PPSMV) using multiple diagnostic tests?
Response 3: We have discussed the available diagnostic tests for detecting PPSMV from samples infected with Sterility Mosaic Disease (SMD). The choice of tests depends on resource availability and affordability, the type of study. Serological assays can detect the virus. Genomic methods, such as sequencing, are useful to identify PPSMV strains accurately.
Comment 4: What are the common practices related to crop rotation, irrigation, and pesticide use?
Response 4: Sterility Mosaic Disease (SMD) is primarily restricted to the Indian subcontinent. In India, pigeon pea crop is cultivated under rain-fed conditions. Since farmers mainly depend on rainfall, only one crop is grown per year, and the land is left fallow until the next crop. During this period, the new flush from stems can provide a habitat for the virus and mite vector to multiply and survive until the next crop season. In addition to that virus and its vector also survive on volunteer pigeonpea plants on bunds or on wild Cajanus species, mainly C. platycarpus and C. scarabaeoides. Crop rotation does not significantly influence the occurrence of SMD. There are a few pesticides available for managing SMD through vector control; however, farmers rarely use them. In India, pigeonpea is mainly cultivated by small and marginal farmers under low-input conditions. Therefore, farmers usually do not spray pesticides for the management of SMD.This is because the infection is not noticeable until symptoms are expressed. Additionally, the mite vector, being very small (200 microns), is not visible to the naked eye, making it challenging for farmers to detect its presence early on and take up pesticide spraying. Farmers mainly rely on resistant varieties for managing SMD.
Comment 5: Are there any other factors that could exacerbate SMD symptoms?
Response 5: The disease occurrence and severity is mainly depends on the primary source of inoculum. Studies show that there is no significant influence of climatic factors in the severity of SMD.
Reviewer 2 Report
Comments and Suggestions for Authors
This is a nice and well orgnized review paper about the pigeonpea sterility mosaic disease (SMD). Here I have few comments for further improving the manuscript:
1, I would say that the SMD is caused by mite-transmitted tenuiviruses instead of PPSMV since several tenuiviruses were found in SMD samples.
2, A figure containing the viral particle, genome strcuture, and phylogeny of PPSMV and PPSMV-2 should be provided in sectioni 2.4.
Minor comments:
1, line 26, ... cuased by mite-transmitted tenuiviruses.
2, line 80. Under certain conditions, it spreads...
3, line 94, please provide the full name of PPSMV before using the abbreviation.
4, line 202, "will be referred to as PPSMV-1 hereafter" I don't agree.
5, line 207, ... was placed in the genus Emaravirus within the family Fimoviridae.
Comments on the Quality of English LanguageThis is a well-written manuscript and is very easy to follow.
Author Response
Comment 1: I would say that the SMD is caused by mite-transmitted tenuiviruses instead of PPSMV since several tenuiviruses were found in SMD samples.
Response 1: While I acknowledge the complexity of the Sterility Mosaic Disease (SMD) etiology, our understanding has significantly advanced since its initial identification in 1999. At that time, the causal agent identified was a mite-transmitted tenuivirus provisionally named Pigeonpea Sterility Mosaic Virus (PPSMV). Subsequent studies have consistently confirmed PPSMV as the primary causal agent. The name PPSMV, recognized by the International Committee on Taxonomy of Viruses (ICTV) and placed under the genus Emaravirus and is widely accepted by the scientific community. This established nomenclature ensures clarity and consistency in scientific communication. Notably, there is an ongoing discussion within the ICTV to adopt binomial nomenclature, proposing PPSMV-1 as Emaravirus cajani and PPSMV-2 as Emaravirus toordali, which may be reflected in future publications. Therefore, we prefer to use of the name PPSMV to describe the primary causal agent of SMD.
Comment 2: A figure containing the viral particle, genome strcuture, and phylogeny of PPSMV and PPSMV-2 should be provided in sectioni 2.4.
Response 2: As suggested, figures are included in the revised manuscript.
Minor comments
Comment A: line 26, ... cuased by mite-transmitted tenuiviruses.
Response A: Since PPSMV placed under the Emaravirus genus, use of PPSMV seems to be more appropriate for the causal agent of SMD disease as most of the researchers in the last decade used the same nomenclature after categorizing it as an Emaravirus rather than Tenuivirus.
Comment B: line 80. Under certain conditions, it spreads...
Response B: The sentence is corrected as per the reviewer’s suggestion and incorporated in the manuscript.
Comment C: line 94, please provide the full name of PPSMV before using the abbreviation.
Response C: Sentence is corrected as per the reviewer’s suggestion
Comment D: line 202, "will be referred to as PPSMV-1 hereafter" I don't agree.
Response D: Sterility Mosaic Disease (SMD) was believed to be caused by a single Emaravirus species known as PPSMV until 2014, when a second Emaravirus associated with SMD was identified. This newly discovered virus was closely related to Fig Mosaic Virus (FMV), another Emaravirus species, rather than the previously identified PPSMV. Consequently, the newly characterized virus was named PPSMV-2, while the initial virus was designated as PPSMV-1 to differentiate between the two. Therefore, referring to the initial virus as PPSMV-1 is necessary for clarity and accuracy in describing the distinct Emaravirus species involved in SMD.
Comment E: line 207, ... was placed in the genus Emaravirus within the family Fimoviridae.
Response E: Sentence is corrected as per the reviewer’s suggestion
Reviewer 3 Report
Comments and Suggestions for Authors
In this review the authors describe the current status of the sterility mosaic disease (SMD) in pigeonpea in India. The authors include all the essential information concerning the history and epidemiology of the disease, the original identification and detection of two emaraviruses associated with SMD, their phylogenetic relationship and the management of the disease mainly via the use of resistant/tolerant pigeopea cultivars. The manuscript is well organized and written and includes all the current literature information. Some minor comments on the English language can be found in the attached file.

Author Response
Comment: In this review the authors describe the current status of the sterility mosaic disease (SMD) in pigeonpea in India. The authors include all the essential information concerning the history and epidemiology of the disease, the original identification and detection of two emaraviruses associated with SMD, their phylogenetic relationship and the management of the disease mainly via the use of resistant/tolerant pigeopea cultivars. The manuscript is well organized and written and includes all the current literature information. Some minor comments on the English language can be found in the attached file.
Response: Thank you very much for your valuable comments and suggestions on our manuscript. We have carefully considered all your comments and have made the necessary corrections in the revised manuscript. Specifically, we have incorporated the minor comments on the English language as suggested in the attached file.